# Severe Course of COVID-19 and Long-COVID-19 in Children: Difficulties in Diagnosis

**DOI:** 10.3390/life13030781

**Published:** 2023-03-14

**Authors:** Elena Vasichkina, Olga Kofeynikova, Svetlana Fetisova, Anastasia Y. Starshinova, Elizaveta Sheyanova, Tatiana Vershinina, Anton Ryzhkov, Aleksey Skripnik, Daria Alekseeva, Elizaveta Nechaeva, Anzhela Glushkova, Dmitry Kudlay, Tatiana Pervunina, Anna Starshinova

**Affiliations:** 1Almazov National Medical Research Centre, St. Petersburg 197341, Russia; 2Medical Department, Saint Petersburg State Pediatric Medical University, St. Petersburg 194100, Russia; 3H.Turner National Medical Research Center for Children’s Orthopedics and Trauma Surgery, St. Petersburg 196603, Russia; 4V.M. Bekhterev National Research Medical Center for Psychiatry and Neurology, St. Petersburg 192019, Russia; 5Pharmacology Department, I.M. Sechenov First Moscow State Medical University, Ministry of Health of the Russian Federation, Moscow 119435, Russia; 6SSC Immunology Institute, FMBA Russia, Moscow 115552, Russia

**Keywords:** children, coronavirus infection, COVID-19, multisystem inflammatory syndrome in children, SARS-CoV-2, long-COVID-19, cardiovascular inflammation

## Abstract

The question of COVID-19 and long-COVID-19 course in children remains unsolved. This infection in children, which is associated with COVID-19, can vary from asymptomatic to systemic damage of various systems. Multisystem inflammatory syndrome in children, associated with SARS-CoV-2 (MIS-C), is a serious condition in children and adolescents after experiencing COVID-19. Published data on MIS-C have indicated that the inflammation can be registered in the gastrointestinal tract (60–100%), as well as in cardiovascular (80%), nervous (29–58%), and respiratory (21–65%) systems. However, with the changing characteristics of SARS-CoV-2, the manifestations of COVID-19 and long-COVID-19 in children have also been changing. Currently, there is no clear understanding of the development of severe COVID-19 and MIS-C in children, especially after being exposed to patients with COVID-19. We presented two new clinical courses of multisystem inflammatory syndrome in children with severe multisystem damage after close contact to relatives with COVID-19 or long-COVID-19. Thus, high-risk children, who are positive for SARS-CoV-2 infection after contact with COVID-19 patients, should be clinically managed during the first few months. The identification of the disease complexity requires the involvement of neurologists, cardiologists, and other specialists.

## 1. Introduction

The global community has been fighting and studying a new infectious disease, coronavirus infection (Coronavirus Disease 2019), caused by a new virus SARS-CoV-2 (severe acute respiratory syndrome coronavirus 2), since December 2019 [1,2]. The acceleration of SARS-CoV-2 transmission led to the registration of a large number of cases of COVID-19 in all countries over in the following months, which necessitated the declaration of a pandemic in March 2020 [3]. Currently, accumulated experience shows that the manifestation of COVID-19 is significantly different in its severity in adults and children [2,4,5].

The presence of various comorbidities [6], age and gender, characteristics of the patient’s immunogenetic status [7], and new strains of SARS-CoV-2 [8,9] have been affecting the manifestation of COVID-19.

In adults, the disease is usually characterized by variability, from asymptomatic forms to acute respiratory failure, cytokine storm, and multisystemic damage with a fatal outcome [5,7,10].

Initially, this infection was thought to be predominantly asymptomatic and mild in children [11,12,13]. However, already by April 2020, British scientists reported about several severe cases of the COVID-19 in children who were characterized by a high temperature 38–40 °C, conjunctivitis, rash, damage to the cardiovascular and respiratory systems as well as the gastrointestinal tract, and the development of the multiple organ dysfunction [14,15,16]. This condition was called multisystem inflammatory syndrome in children, associated with SARS-CoV-2 (MIS-C) [17,18]. More than 2600 cases of MIS-C were reported in 2020, according to published data of the Centers for Disease Control and Prevention. At the end of June 2022, 8639 cases of this syndrome and 70 deaths were registered in the USA [18,19,20].

A multicenter study was carried out in the Russian Federation. It described results where 245 children with MIS-C were treated. Their age ranged from 3 months to 17 years [21]. It was shown that patients had damage to the cardiovascular system (66%), as well as gastrointestinal disorders (61%), every fourth patient had neurological disorders (27%) and signs of respiratory failure (19%).

There is an interesting fact that MIS-C can develop in patients with asymptomatic COVID-19 and patients with severe disease [22,23,24,25]. It is not clear which factors are predisposing to the development of MIS-C after COVID-19, as well as examination methods that should be used for the diagnosis of MIS-C [26,27]. In addition, despite numerous studies, the pathophysiological mechanism of the development of this syndrome and the entire spectrum of clinical manifestations, risk factors for the severe course of the disease, and the range of necessary examinations are not fully understood.

Long-COVID-19, caused by SARS-CoV-2, is the second long-term infection that can lead to the development of polyorganic disorders [22]. The characteristics of long-COVID-19 are mainly studied in adult patients.

According to the current data, the course of long-COVID-19 and specific symptoms in children are still the subject to the study. Some studies have presented that long-COVID-19 is characterized by damage to the respiratory system (21.2%), nervous disorders (16%), skin (15%), the gastrointestinal tract (13%), cardiovascular system (11%), as well as possible psychiatric symptoms (10%), and pathologies of the intestinal (9%) system [23].

In addition, MIS-C and long-COVID-19 are also characterized by heterogeneous and multisystemic lesions, while progressing variability is noted from simple subjective symptoms. These variety symptoms certainly have difficulties in differential diagnosis [24].

In the article, the authors present two clinical cases that demonstrate difficulties in the diagnosis of polyorganic disorders in children who have been infected by SARS-CoV-2 during different periods of the disease. Our experience in managing of these patients demonstrates the development of polyorganic disorders in children with COVID-19 and long-COVID-19, as well as challenges in differential diagnosis that require the exclusion of a wide range of systemic and infectious diseases.

## 2. Case 1

A patient aged 12 was admitted to the Department of Pediatric Cardiology for a comprehensive cardiological examination and for specification of the diagnosis specification due to the septal leaflet in the tricuspid valve. This disorder was detected on the 15th day after testing positive for COVID-19. There were no complaints.

According to the medical data, in April 2022, the disease acutely began with persistent fever (39–40 °C) that lasted 9 days, decreased appetite, abdominal pain, and semi-liquid feces. Symptomatic therapy was prescribed (Ibuprofen, Enterol).

By the fourth day, when walking, pain worsened in the left ankle.

On the sixth day, the patient’s stool returned to normal; however, complaints about fever and joint pain persisted.

On the eighth day, the child was hospitalized. The condition was assessed as moderate due to intoxication syndrome. Upon examination, the body temperature increased to 38.5 °C, the skin was without rash, peripheral lymph nodes were not enlarged, and there was shortness of breath on auscultation in the left side. There was an increase in the volume of the left ankle (S > D by 2 cm), limitation of mobility, and moderate pain. Signs of arthritis were revealed with the use of ultrasound of the ankle joint.

Three weeks before the appearance of complaints, the boy was in contact with a patient who had COVID-19, the PCR test for SARS-CoV-2 (upon admission) was negative, but IgG to SARS-CoV-2 was detected.

### 2.1. Examination Data

According to the laboratory data, we found thrombocytopenia up to 107 × 10^9^/L, a significant increase in the level of C-reactive protein (CRP) up to 122.6 mg/L and procalcitonin up to 33 ng/mL (normal up to 2), ferritin up to 377 mcg/L (normal up to 140), fibrinogen up to 5.04 g/L, D-dimer up to 3.24 µg/L (normal up to 0.56), and hypoalbuminemia up to 20 g/L.

Left-sided segmental (S4) pneumonia was diagnosed by X-ray examination.

Previously, the child was prescribed antibacterial and symptomatic therapy with low molecular weight heparins.

During the examination in the hospital, the boy had a body temperature up to 38 °C, the skin was without rash, there was no hemorrhagic syndrome, and the peripheral lymph nodes were not enlarged and were painless. Heart sounds were sonorous, heart rate was 76 beats per minute, and blood pressure was 100/60 mm Hg. When auscultation, breathing was weakened and wheezing was not heard. The abdomen was soft and painless, and the liver and spleen were not enlarged.

Echocardiography (ECG) and 24 h ECG monitoring revealed no rhythm and conduction disturbances, and no ischemic changes were registered (Figure 1).

Applying echocardiography data, prolapse in the diastole cavity of the right ventricle was revealed. The projection of the tricuspid valve leaflets, a homogeneous formation with fuzzy contours, 16 × 6.5 mm in size, on a wide base were examined (Figure 1A).

The dimensions of the heart were not enlarged, and the contractility of the right and left ventricles were preserved. The condition was regarded as the course of infective endocarditis of an unspecified etiology with tricuspid valve damage.

During echocardiography, the formation of a tricuspid septal leaflet on a wide base, and a thrombus could not be ruled out. It was clearly visualized with even edges (12.5 × 7 mm in size) and a tricuspid septal leaflet was not floating. The biomechanics of the valve were not impacted. The dimensions of the heart chambers were normal and myocardial contractility was preserved; however, basal lower septal segment dyskinesia was detected (Figure 1B).

A formation with clear contours was determined. There were no signs of myocardial edema. On delayed post-contrast images, there was no distinct pathological accumulation of the contrast agent in the myocardium (Figure 2).

Chest computed tomography (CT) was performed with the use of a contrast agent (Figure 3).

The CT and CT angiography examined areas of increased density of the ground glass type (GGO); defects in contrast with the distal branches of A8 of the right lung (PE) were diagnosed. In the anteromedial sections of the right atrium, at the level of the tricuspid valve, an extended avascular hypodense area of an irregular shape was determined (up to 16.5 × 7.5 × 9 mm in size).

For differential diagnosis and exclusion of heart formation, positron emission tomography was performed—chest CT: fluorodeoxyglucose—positive neoplasms in the projection of heart structures (including the right atrium) were not detected.

The patient was consulted by a rheumatologist, and systemic autoimmune diseases were excluded.

### 2.2. Substantiation of Diagnosis

Taking into account the patient’s age, epidemiological data (contact with COVID-19); detection of IgG to SARS-CoV-2; duration of fever; the elevation of enzymes in the blood, hypoalbuminemia; a significant increase in CRP, ferritin, and D-dimer; multiple organ damage with signs of gastrointestinal tract involvement (abdominal syndrome), abnormalities of the blood coagulation system; and segmental viral pneumonia, a diagnosis of a multisystem inflammatory syndrome associated with COVID-19 was determined.

During the hospitalization, antibiotic therapy was completed, and the anticoagulant therapy was continued. The patient received glucocorticoid therapy for seven days. In the period of hospitalization, there were positive treatment dynamics in the patient: normalization of body temperature, relief of intoxication syndrome, no signs of arthritis of the left ankle, and normalization of the level of inflammatory markers—CRP, ferritin, fibrinogen, transaminases, albumin, and platelets. The patient was discharged in a fair condition, with a good prognosis under supervision, along with warfarin administration.

## 3. Case 2

A patient aged 7 was hospitalized in January 2022 in critical condition with complaints of shortness of breath, left hand and left leg weakness, and facial asymmetry.

In the end of December 2021, the girl suffered from a respiratory infection with a fever within 3 days after contact with her mother who had COVID-19. IgM and IgG of SARS-CoV-2 were found. The girl was not examined. The PCR test for SARS-CoV-2 was negative

Pain in the epigastric region appeared within two weeks after recovery. There was no vomiting, nausea, or episodes of fever. Dyspnea at rest and tachycardia appeared two days later. The patient was urgently hospitalized. The condition upon admission was regarded as severe due to cardiac and respiratory failure.

Concerning the laboratory data, the level of enzymes was increased: creatine kinase of 311 U/L (normal up to 154), lactate dehydrogenase of 715.4 U/L (normal up to 295), and AST of 76 U/L (normal up to 51). The AST level increase was registered first and was not associated with the child’s condition or with the therapy. Hepatitis was excluded.

Erythrocyte sedimentation (ESR) rates up to 20 mm/h (2–10) and the level of ferritin and procalcitonin revealed hypoalbuminemia up to 20 g/L (35–50) were determined.

Furthermore, after increasing troponin I to 0.0410 ng/mL (0.0000–0.0160), N-terminal prohormone of the brain natriuretic peptide to 11.115 pg/mL (normal up to 145.00), and D-dimer to 3.32 μg/kg (normal 0.1–0.56), hypoalbuminemia was observed. At the same time, the level of C-reactive protein (CRP) was 9 mg/L, and ESR, ferritin was in a normal condition. Differential diagnosis was carried out with other viral infections, such as cytomegalovirus infection (CMV); herpes types 1, 2, and 6; Epstein−Barr virus; and parvovirus B19. The results of PCR and immunologic data with antigens were negative.

The data of the chest computed tomography (CT), including contrast agent for determination of vascular thrombosis, are shown in Figure 4.

The CT and CT-angiography, a diffuse decrease of the lung pneumatization was diagnosed due to extensive confluent “ground-glass” attenuation, and free fluid up to 25 and 30 mm was determined in the pleural cavities on the right and left, respectively. The signs of left-sided thromboembolism of subsegmental branches of the A8 pulmonary artery and heterogeneity of pneumatization were diagnosed (Figure 4).

Sinus tachycardia was recorded on an ECG with a heart rate of 125 per minute. EOS deviation to the right was also noted. The ST segment was in depression in II, III, aVF, and V4–V6 up to 2.5 mm.

Echocardiography revealed dilatation of the right ventricle up to 36 mm (z-score 2.11), dilatation of pulmonary artery trunk 26 mm (Boston z-score 2.89), and dilatation of the left atrium up to 34 mm (Boston z-score 3.11) [20]. Decreased contractility of the left ventricle to 43% (Teicholz), increased pressure in the pulmonary artery to 57 mm Hg, and tricuspid valve insufficiency of the second degree were noted. The diameter and walls of the coronary arteries were not changed. The phenomenon of “spontaneous contrast” was noted.

Antibiotic therapy with amoxicillin, antiviral therapy with umifenovir, and multicomponent therapy for heart failure (captopril, spironolactone, and furosemide) were prescribed.

On the second day, neurological symptoms with complaints of slurred speech, left-sided hemiparesis, and paresis of mimic muscles in the central type on the left were registered.

Based on the brain MRI in the lumen of M1 of the right middle cerebral artery (MCA), a thrombus 12 mm long with an ischemic zone in the subcortical structures on the right, severe hypoperfusion of the right fronto−parietal−temporal region of the brain was detected. The patient’s condition was regarded as acute cerebrovascular accident (ACV) in the territory of the right middle cerebral artery according to the ischemic type, left-sided hemiparesis according to the central type, and central type of dysfunction of the hypoglossal nerve; in addition, facial nerve deficit was noted.

The child received anticoagulant therapy with heparin for three days and then changed to low molecular weight heparin therapy for a long time before thrombus lysis.

The developed multiorgan pathology was differentiated from systemic auto-inflammatory diseases. An antinuclear factor, antibodies to cardiolipin IgM and IgG, lupus anticoagulant, and the levels of proteins S and C were normal.

In addition, blood diseases with the pathology of the coagulation system and/or platelets were excluded. According to the results of a genetic examination, the main thrombophilic mutations and abnormalities of the folate cycle were not found. During the examination, the girl also underwent blood sampling for whole genome sequencing, which did not reveal any mutations.

Signs of the signal enhancement on Tirm due to edema of the IVS myocardium were revealed on the heart MRI data with contrast, which indicated inflammatory changes in the myocardium three weeks later (Figure 5 and Figure 6).

From the ECG, there was also a positive trend in the form of improved repolarization processes.

From the brain MRI, there was restoration of patency of the right MCA with an MR picture of postischemic cystic−gliotic changes in the territory of the right middle cerebral artery observed (Figure 7).

We diagnosed long-COVID-19, taking into account the age of the patient; epidemiological data (contact with parents with COVID-19 and positive IgM and IgG to SARS-CoV-2); viral pneumonia; and the development of severe multisystem organ pathology with involvement of the gastrointestinal tract (abdominal syndrome), heart (with the development of acute myocarditis), and CNS (ischemic stroke in the territory of the right MCA) that appeared after the onset of the disease.

During the period of hospitalization, there was a clear positive trend in the patient: normalization of the level of troponin I and a decrease in NT-proBNP, normalization of the myocardial contractility left ventricular, and improvement of the regional contractility. There was an improvement in muscle strength in the extremities and an increase in motion, positive dynamics of the neurological status, and thrombus lysis in the MCA, as confirmed by brain MRI.

The child began to roll over in both directions, sit down, get up, and walk independently.

The patient was discharged in a satisfactory condition under supervision with warfarin, acetylsalicylic acid, and omeprazole administration.

## 4. Discussion

According to the published data, the incidence of MIS-C is 1:4000 children who had COVID-19 infection [28].

The severity of the course and prognosis are determined by the development of hypotension/shock symptoms. The degree of myocardial damage and neurological disorders have been mostly revealed [29,30].

This article presents the heart and brain variants associated with COVID-19. The specificity of these cases was that the clinical and laboratory data did not fully fit the combinations we found in the literature data [3,15,16].

Thus, our patients had no fever associated with the development of multiorgan failure syndrome according to the WHO, the Center for Disease Control and Prevention (The USE)/, and the Royal College of Pediatrics and Child Health (The UK) recommendations [3,15]. The laboratory data, such as ESR, CRP, procalcitonin, and ferritin, were not increased.

In the process of differential diagnosis, various diseases were excluded, including systemic autoinflammatory diseases, blood diseases with the pathology of the coagulation system and/or platelets, and cardiomyopathy. Nevertheless, we diagnosed multisystem inflammatory syndrome associated with COVID-19.

We introduced a comparative analysis of characteristics for serious conditions developed in children after being infected by SARS-CoV-2, as shown in Table 1.

In clinical case 2, the patient had all six MIS-C criteria, according to the WHO. Clinical case 1 met only four of them.

In 2023, the new criteria for the multisystem inflammatory syndrome in children CSTE/CDC MIS-C were published [19]. Applying these criteria, the first clinical case fit this syndrome, while the second case remained uncertain regarding the presence of MIS-C in the patient.

Such multisystem pathology could be caused after COVID-19 and also after being infected by SARS-2-CoV-2 and long-COVID-19, with the persistence of the virus without the capacity for detection by routine laboratory diagnostic methods [31].

Currently, the period of multisystem disorders varies after suffering from COVID-19. According to the WHO, some symptoms can be expected in 3 or more months, and the clinical picture is from 4 to 12 weeks [30,31].

Moreover, with the accumulation of experience, there is a necessity to develop additional criteria for diagnosed lesions of the cardiovascular and nervous system both in the structure of MIS-C and in the period of observation of children in the future [19,31].

Recent studies have shown that patients with MIS-C have an extended variability in the TCR b 11-2 (TRBV11-2) gene and cytokine levels that are correlated with the severity of the clinical manifestations and course of this syndrome. Analysis of the sequences of the SARS-CoV-2 additionally revealed the cause of the functional impairment of neuronal cells with the development of a neurotoxic reaction, which contributed to the appearance of severe neurological and cardiological symptoms [32].

This article presents a clinical variant of severe MIS-C in a child who became infected with COVID-19 after contact with their sick mother. After the classical manifestations of a viral infection, the development of acute myocarditis, ischemic stroke, and pulmonary embolism were noted.

Currently, predictors of MIS-C have not identified. Subsequent studies will likely identify a group of children at high risk of developing this serious condition against the background of exposure to SARS-CoV-2.

Some studies indicate that chronic pathology, bronchial asthma, diabetes mellitus, obesity, and immunosuppression might be risk factors in the development of MIS-C [32,33]. Local studies have confirmed the relationship between severe MIS-C in children with obesity [18]. Certain difficulties are presented by the proof of an unambiguous connection between the development of multisystem inflammation and SARS-CoV-2.

In adult patients with COVID-19 with myocarditis, positive first and second-level test results were obtained in 23% of cases [33]. In pediatric practice, it remains problematic to obtain the presence of clinical manifestations of a viral infection during the period of a rise in viral activity or upon contact with a patient with COVID-19 [2,12]. According to the results of the examination of children with MIS-C in domestic practice, positive PCR tests were obtained in 44% of cases [18].

In international practice, with the development of characteristic symptoms with changes in computed tomography of the chest during this pandemic, even without a positive PCR test, the disease is regarded as COVID-19 [34].

The presented clinical cases do not allow for unambiguous confirmation of the signs that are characteristic of COVID-19, described in the literature, which unambiguously diagnose this complication [29,30,34].

Earlier studies have noted that the syndrome most often develops in children of the African/African American (AFR) population [35].

Previously, it has been shown that a severe course of COVID-19 can be associated with biomarker predictors—genes, associated with immune response (HLA genotype, genes coding cytokines, and proinflammatory mediators receptors) and specifics in ACE-2 expression and its transporter system proteins (SLC6A20, TMPRSS2 genes). The review shows that severe biomarkers were detected more commonly in men [7]. Russian scientists have noted that in boys, the course of MIS-C was also the most severe in 57% of cases. The syndrome was diagnosed in 64% of children of the Eastern European population, but there was no correlation with the severity of MIS-C [21]. The conducted studies showed the significance of an increase in ESR in 69%, procalcitonin in 69%, ferritin in 77%, and CRP in 98% [18].

The issue of differential diagnosis of MIS-C with other pathological conditions, including other infectious diseases that occur with similar clinical and laboratory symptoms, is extremely relevant [36,37].

## 5. Conclusions

During the COVID-19 pandemic, the global community faced a large number of new manifestations and symptoms. One of which being multisystem inflammatory syndrome. This severe pathological condition, associated with COVID-19 in children, is attracting more and more attention from specialists. Today, understanding of the risk factors, predictors and causes of development, variability in the course, and long-term prognosis of MIS-C is under study.

Relatively favorable outcomes have been reported for this syndrome, which have led to death in 2% of cases, taking into account the timely diagnosis and effective treatment so far. However, the consequences of MIS-C, especially concerning the development of cardiac, neurological, and autoimmune disorders in children and adolescents, remain unknown. It should be taken into account that the course of this condition may change simultaneously with the emergence of new strains of SARS-CoV-2.

Information about the past infection or the data considering contact with a patient with COVID-19 in a child during the first few months should warrant an in-depth examination in a hospital with the involvement of neurologists, cardiologists, radiology specialists, etc. Children who have been infected with COVID-19 require further follow-up. Patients who have undergone MIS-C deserve special attention as the long-term results of the course of this syndrome were not known before.

The course of COVID-19 has undergone significant changes in adults and children since the emergence of the infection in the world. Perhaps, in the future, specialists will have to deal with new variants of the course of MIS-C and the consequences of COVID-19 in pediatric practice. Currnently, knowledge on the pathogenesis, diagnosis, treatment, and prevention of long-COVID-19 in children is still lacking [37]. Clinical symptoms and the involvement of various organs and systems, including cardiovascular, in long-COVID-19 in the pediatric population requires study in order to understand these issues better.

## Figures and Tables

**Figure 1 life-13-00781-f001:**
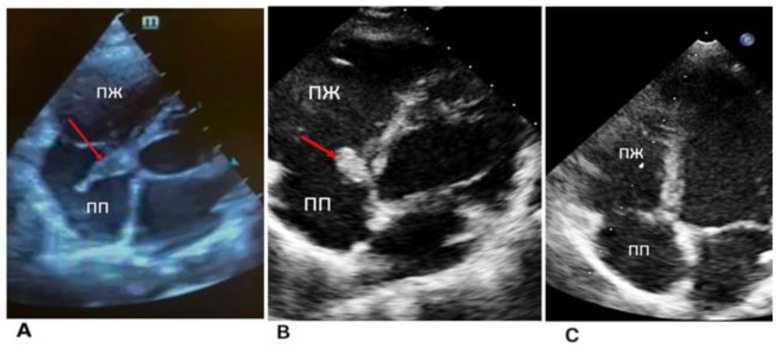
Echocardiography (dynamics of thrombus regression). (**A**) Thrombus was in the septal leaflet of the tricuspid valve (red arrow) upon admission, size 16 × 6.5 mm. (**B**) Two weeks after the start of anticoagulant therapy. The red arrow indicates a thrombus with dimensions of 12.5 × 7 mm. (**C**) Lysis of a thrombus in the septal leaflet.

**Figure 2 life-13-00781-f002:**
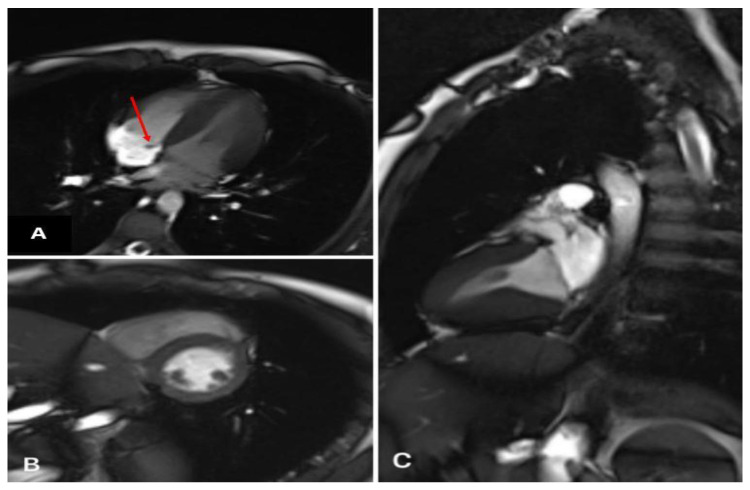
Heart MRI with contrast. (**A**) Four-chamber plane (formation of a septal valve—red arrow). (**B**) Two-chamber plane along the short axis (normal). (**C**) Two-chamber plane along the long axis (norm).

**Figure 3 life-13-00781-f003:**
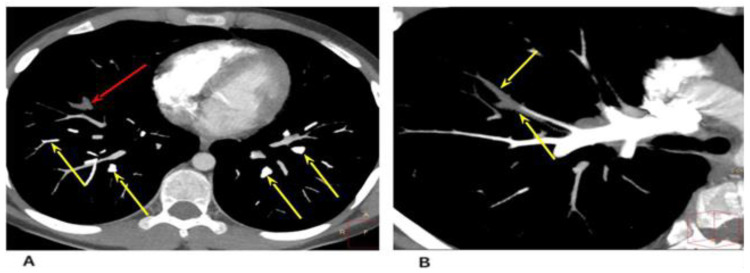
Cardiac CT angiography, axial plane, maximum intensity projection (MIP), vascular mode. (**A**) Contrasted distal branches of the pulmonary artery (yellow arrows). Contrasting defects in the distal branches of the A8 right lung (PE—red arrow). (**B**) Contrasting defects of the distal branches of A8 of the right lung (PE).

**Figure 4 life-13-00781-f004:**
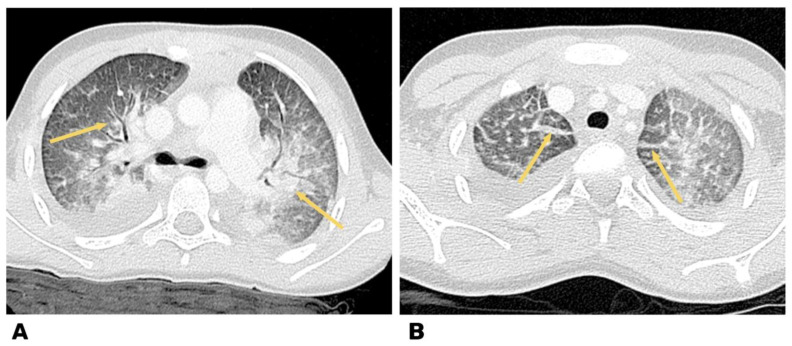
Heart CT angiography, axial plane, and pulmonary regimen in a 7-year-old child. (**A**) Reduction in lung pneumatization due to extensive confluent ground glass attenuation. (**B**) Free fluid was determined in the pleural cavities on both sides.

**Figure 5 life-13-00781-f005:**
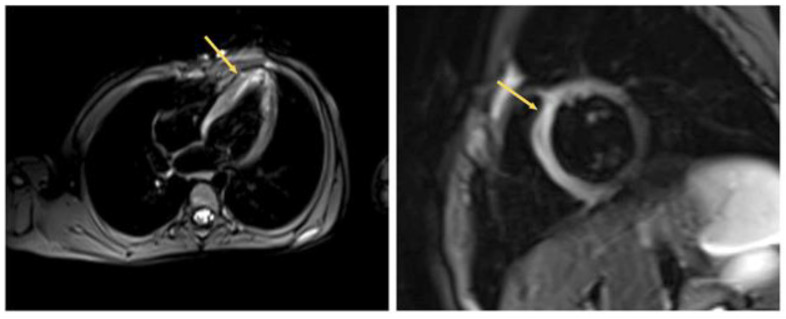
Heart MRI with contrast. Arrows indicate myocardial edema in the region of the interventricular septum.

**Figure 6 life-13-00781-f006:**
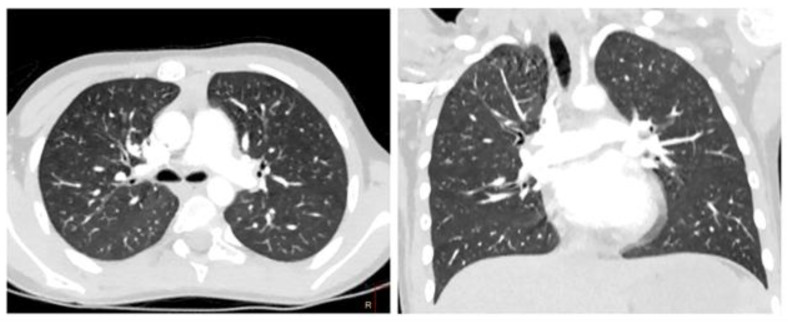
Heart CT, axial and coronal angiography, and pulmonary mode after 4 weeks.

**Figure 7 life-13-00781-f007:**
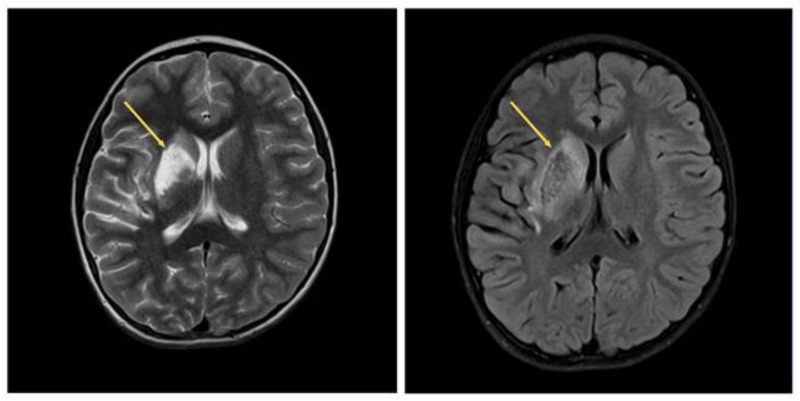
Brain MRI. Arrows indicate postischemic cystic−glial changes in the territory of the right middle cerebral artery.

**Table 1 life-13-00781-t001:** Comparison of characteristics of serious condition developed in children after COVID-19.

WHO Criteria for MIS-C	Whittaker E, 2020 [24]	Kaushik, 2020 [30]	Feldstein, 2020 [31]	Clinical Case 1	Clinical Case 2
	(%, Number of Patients with Symptom/Total Number of Patients)				
1. Age 0–19 years	100% (58/58)	100% (33/33)	no data	+	+
2. Fever ≥ 3 days	100% (58/58)	93.9% (31/33)	no data	+	-
3. Clinical signs of a multisystem involvement (at least 2):	100% (58/58)	100% (33/33)	no data	+	+
Rash, bilateral nonpurulent conjunctivitis, or signs of inflammation of the skin or mucous membranes (mouth, hands, or feet)	rash—51.7% (30/58) conjunctivitis—44.8% (26/58) lesion of mucous membranes—29.3% (17/58)	100% (33/33)	bilateral non-purulent conjunctivitis—55.3% (103/186) rash—59.2% (110/186)	+	+
Hypotension or shock	50% (29/58)	63.6% 21/33	no data	-	-
Cardiac dysfunction, pericarditis, valvulitis, or coronary abnormalities (include ECHO signs or elevated levels of troponin/BNP)	coronary artery’s aneurysm 13.7% (8/58) dysfunction LV—62% (18/29) elevation of troponin—68% (34/50) elevation of BNP —82.7% (24/29)	elevation of BNP—50% (16/32) NT-proBNP 16 pericarditis—45.5% (15/33) decrease EF LV < 30%—12.5% (4/32) EF LV 30–50% —53.1% (17/32)ectasia of the coronary artery—6.7% (2/32)	cardiac dysfunction, pericarditis—80.1% (149/186) elevation of troponin—50.3% (77/153) elevation of BNP—73.4% (94/128)coronary artery dilatation—8.1% (15/186)	-	+
Signs of coagulopathy (prolongation of prothrombin time, increased levels of D-dimer)	signs of coagulopathy 100% (58/58)	signs of coagulopathy 100% (58/58)	increased levels of D-dimer—66.9% (79/118)	+	+
Acute gastrointestinal symptoms (diarrhea, vomiting, or abdominal pain)	vomiting—44.8% (26/58), abdominal pain—53.4% (31/58) diarrhea—51.7% (30/58)	acute gastrointestinal symptoms 100% (33/33)	acute gastrointestinal symptoms—91.9% (171/186)	+	+
4. Elevated levels of inflammatory markers (for example, ESR, CRP, or procalcitonin)	100% (58/58)	100% (33/33)	no data	+	+
5. Absence of other known inflammatory reasons, including bacterial sepsis and staphylococcal/streptococcal toxic shock syndrome	100% (58/58)	100% (33/33)	no data	+	+
6. Evidence of SARS-CoV-2 infection	77.5% (45/58)	no data	no data	+	+
Positive SARS-CoV-2 PCR	25.8% (15/58)	no data	no data	-	-
Positive serology tests	86.9% (40/46)	no data	no data	+	+
Positive antigen test	no data	no data	no data	-	-
Epidemiological data (patient’s contact with COVID-19-positive subject)	n/a	no data	no data	+	+

Explanations; +—criterion presence; -—criterion absence.

## Data Availability

Not applicable.

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
