# Peer review of "Severe Course of COVID-19 and Long-COVID-19 in Children: Difficulties in Diagnosis"

_life, 2023, doi:10.3390/life13030781_

Round 1

Reviewer 1 Report

The title is informative and relevant. The references are relevant and recent. The cited sources are referenced correctly. Appropriate and key studies are included. The introduction reveals what is already known about this topic. The research question is clearly outlined.

The case reports are well-described, the used methods methods for diagnosing and therapy are valid and reliable. The patient data are presented in an appropriate way. The illustrative materials are relevant and clearly presented. Table 1 is comprehensive and interesting approach of presenting and comparing data from the other published papers.

Data is discussed from different angles and placed into context without being overinterpreted.

The conclusions are supported by references and own results.

This paper added to what is already in the topic. The article is consistent within itself.

Specific comments on weaknesses of the article and what could be improved:

Major points 

1. It is not clear why these particular cases are chosen for the publication. The authors should explain this in the end of the introduction.

2. No limitations and strengths of the research are presented.

3. It would be beneficial for the paper to give some recommendations based on your observations.

Minor

1. page 1, line 24 - "In this report, the authors pointed out the change in the parameters for...", it should be "..., we pointed out..."

2. minor spelling and grammar - for example - lack of full stop in line 43/page 1, etc.; line 107 - starting the sentence with number, 109 - extra space, etc.

3. page 2/line 79 - what is the meaning of "It should be noted that according to the ECG 79 data, changes were noted only in 4% of cases."

Author Response

Dear colleague,

Thank you very much for your comments and recommendations.

Follow corrections:

  1. It is not clear why these particular cases are chosen for the publication. The authors should explain this in the end of the introduction. – was corrected
  2. No limitations and strengths of the research are presented. – was corrected
  3. It would be beneficial for the paper to give some recommendations based on your observations. – was corrected

Minor

  1. page 1, line 24 - "In this report, the authors pointed out the change in the parameters for...", it should be "..., we pointed out..." – was corrected
  2. minor spelling and grammar - for example - lack of full stop in line 43/page 1, etc.; line 107 - starting the sentence with number, 109 - extra space, etc. – was corrected
  3. page 2/line 79 - what is the meaning of "It should be noted that according to the ECG 79 data, changes were noted only in 4% of cases." – was corrected

Reviewer 2 Report

In the submitted paper, the authors present two cases of MIS-C in children. The reviewer's doubts are raised by the diagnosis of the MIS-C disease. The authors write about the lack of unequivocal criteria for diagnosis; meanwhile, such criteria have been set by both the CDC and the American Academy of Pediatrics. (https://www.cdc.gov/mis/mis-c/hcp/index.html; https://www.aap.org/en/pages/2019-novel-coronavirus-covid-19-infections/clinical-guidance/multisystem-inflammatory-syndrome-in-children-mis-c-interim-guidance/). (https://www.thelancet.com/action/showPdf?pii=S2352-4642%2820%2930304-7).

The authors recognize MIS-C without taking into account generally accepted diagnostic criteria. The basic diagnostic criterion is fever, the importance of which has been reduced in this article. Also, the currently recommended treatment was not undertaken. Can MISC-C be diagnosed in described patients, or they were the cases of COVID-related thrombosis? In the description, the authors use the treatment typical for thrombosis. The authors do not mention the currently generally recommended treatment.

The article is written in very poor English. After a nice introduction, the case reports contain a number of errors: unexplained abbreviations, colloquial phrases, repetitions. There are so many errors that the reviewer gave up correcting them. Both case reports need improvement. For the clarity of the article, abbreviations should be explained the first time they are used.

The non-uniform description of cases is completely unclear. The layout of the first description differs from the second, as written by two people who did not communicate; in the second there are subsections which were not included in the first.

Below there are examples of errors:

Line 105/106 - "contact with ill COVID-19 105 mother ille " it is probably mistake;

line 112 - " cretinkinase " ?; rather creatine kinase

line 113 - AST - shortage should be explained

line 114- NT-proBNP - shortage should be explained

line 119 " The data of the chest computed tomography (CT), including with contrast agent fordetermination vascular thrombosis, are shown in Figure 1." without including

line 126: "a diffuse decrease of the lung pneumatiza- 125 tion was diagnosed due to extensive confluent ground glass attenuation " In the opinion of the reviewer, this is an incorrect notation: lung pneumatization is not caused by of the ground glass attenuation.

line 131 - shortages should be explained

line 144 "asynergy in the left extremities, and facial asymmetry" I do not understand, what the authors mean. What is asynrgy in left extremity? Paralysis of facial nerve? I looked below, but it is not described.

line 152 "Heparin anticoagulant therapy with the low molecular weight heparin therapy with dalteparin sodium were prescribed " The sentence should be improved

line 161 "3 weeks after signs of the signal enhancemen on Tirm due to edema " I think TIRM is the dark fluid sequence. It shoulb be described

line 169 "Against the background of " ?

line 173 "activity and muscle strength in the left extremities were ". I think the patient had one lefy extremity.

line 178 "period of hospitalization, the patient showed a clear positive trend: normalization of the level of Troponin " Why the name troponin is written with big letter?

line 190 "leaflet of the tricuspid " What the authors mean?

line 195 "the child became pain intensifying when walking in the left ankle " The sentence should be improved

line 200 " the skin was clean " ? Is it correct?

line 201 "Weakening of breathing on the left side was revealed by auscultation. " The sentence should be improved

line 209-215 The word " according" is repeated three times.

line 224 " the skin was clean " ? Is it correct?

line 242, line 249 - Why do authors start all paragraphs with the word "according"?

line 297- as above

line 309 _ shortage should be explained

line 317 "the child had not previously been seen " colloquial phrase

line 336-337 repetitions

line 362 "There are currently no standard criteria for defining MIS-C. " As above

The record of the literature is variable. Sometimes the authors added underlined DOI number, sometimes they didn't. References need improvement. 

The authors have prepared two abstracts for the article. Please clarify whether the journal allows such an arrangement. The reviewer is not aware of this requirement.

In my opinion, the article requires very thorough work, language correction, reference to recognized, current recommendations

Author Response

Dear colleague,

Thank you very much for your comments and recommendations.

Follow corrections:

Line 105/106 - "contact with ill COVID-19 105 mother ille " it is probably mistake – was corrected

line 112 - " cretinkinase " ?; rather creatine kinase - was corrected

line 113 - AST - shortage should be explained – was explained

line 114- NT-proBNP - shortage should be explained – was explained

line 119 " The data of the chest computed tomography (CT), including with contrast agent fordetermination vascular thrombosis, are shown in Figure 1." without including  -  was corrected

line 131 - shortages should be explained –  information was included

line 144 "asynergy in the left extremities, and facial asymmetry" I do not understand, what the authors mean. What is asynrgy in left extremity? Paralysis of facial nerve? I looked below, but it is not described. – was corrected

line 152 "Heparin anticoagulant therapy with the low molecular weight heparin therapy with dalteparin sodium were prescribed " The sentence should be improved – was corrected

line 161 "3 weeks after signs of the signal enhancemen on Tirm due to edema " I think TIRM is the dark fluid sequence. It shoulb be described  - was corrected

line 169 "Against the background of " ? - was corrected

line 173 "activity and muscle strength in the left extremities were ". I think the patient had one lefy extremity. - was corrected

line 178 "period of hospitalization, the patient showed a clear positive trend: normalization of the level of Troponin " Why the name troponin is written with big letter? - was corrected

line 190 "leaflet of the tricuspid " What the authors mean? – it is «the septal leaflet in tricuspid valve» - was corrected

line 195 "the child became pain intensifying when walking in the left ankle " The sentence should be improved –  was corrected

ine 200 " the skin was clean " ? Is it correct?  - was corrected

line 201 "Weakening of breathing on the left side was revealed by auscultation. " The sentence should be improved - was corrected

line 209-215 The word " according" is repeated three times. – was corrected

line 224 " the skin was clean " ? Is it correct? – was corrected

line 242, line 249 - Why do authors start all paragraphs with the word "according"? – was corrected

line 297- as above - was corrected

line 309 _ shortage should be explained - - was corrected

line 317 "the child had not previously been seen " colloquial phrase - was corrected

line 336-337 repetitions – was corrected

line 362 "There are currently no standard criteria for defining MIS-C. " As above - was corrected

The record of the literature is variable. Sometimes the authors added underlined DOI number, sometimes they didn't. References need improvement. – was corrected

The authors have prepared two abstracts for the article. – was corrected

Round 2

Reviewer 1 Report

The paper has been improved significantly.

Minor comments:

1. There is a misspelling in the title - "Lond-COVID".

2. "2.1."- There is no title there

Author Response

Dear colleague,

Thank you very much for your recommendations.

Follow corrections:

  1. There is a misspelling in the title - "Lond-COVID". – was corrected
  2. "2.1."- There is no title there – was corrected

Reviewer 2 Report

The authors have significantly improved the content of the article. The job title has been changed. However, there are still minor remarks.

line 27 : mistake

line 32: The dates should not be infected by Sars-CoV2: change the meaning

line 105: meaning of the sentence is not correct. Rather: By the 4th day of the disease, the pain in the left ankle got worse when walking.

line 336: mistake

lines 305-312 and line  345-348 contain the same text. Why?

line 328 - In the text, the authors inform about the new criteria for MIS-C recognition; meanwhile, the reference moves to the 2021 article. It's not compatible.

Author Response

Dear colleague,

Thank you very much for your recommendations.

Follow corrections:

line 27 : mistake – was corrected

line 32: The dates should not be infected by Sars-CoV2: change the meaning – was corrected

line 105: meaning of the sentence is not correct. Rather: By the 4th day of the disease, the pain in the left ankle got worse when walking. – was corrected

line 336: mistake – was corrected

lines 305-312 and line  345-348 contain the same text. Why? – was corrected

line 328 - In the text, the authors inform about the new criteria for MIS-C recognition; meanwhile, the reference moves to the 2021 article. It's not compatible. -  was corrected
